# Tunable Waveguides Couplers Based on HPDLC for See-Through Applications

**DOI:** 10.3390/polym13111858

**Published:** 2021-06-03

**Authors:** Sergi Gallego, Daniel Puerto, Marta Morales-Vidal, Manuel G. Ramirez, Soumia I. Taleb, Antonio Hernández, Manuel Ortuño, Cristian Neipp

**Affiliations:** Instituto Universitario de Física Aplicada a las Ciencias y las Tecnologías, Universidad de Alicante, Apartado 99, 03080 Alicante, Spain; dan.puerto@ua.es (D.P.); marta.morales@ua.es (M.M.-V.); ramirez@ua.es (M.G.R.); cristian@gcloud.ua.es (S.I.T.); ahernandez@ua.es (A.H.); mos@ua.es (M.O.); cristian@ua.es (C.N.)

**Keywords:** holography, holographic recording materials, photopolymers, waveguides, see-through

## Abstract

Photopolymers have become an important recording material for many applications, mainly related to holography. Their flexibility to change the chemical composition together with the optical properties made them a versatile holographic recording material. The introduction of liquid crystal molecules in a photopolymer based on multifunctional monomer provides us the possibility to generate tunable holograms. The switchable holographic elements are a key point for see-through applications. In this work, we optimize the holographic polymer-dispersed liquid crystals composition to improve the performance of tunable waveguide couplers based on transmission gratings and specifically their response under an applied electric field. A variation around 60% in the transmission efficiency was achieved.

## 1. Introduction

The work to obtain a thin, light, aesthetically pleasing pair of augmented reality (AR) glasses is in progress. Currently, in the design of smart glasses some problems remain on the table, like power consumption, the limitation of the resolution, the wide field of view, etc. [1,2,3]. There are different steps to fix in order to fabricate a smart glass: sensing, processing, image generation, the transformation of an analogical image to a digital one, propagation through the glass to the eye and the generation of the virtual image on the retina. In the last two processes, the inclusion of a holographic optical element has provided some possible solutions. In particular, photopolymers have been reported as a good system to bring the photons produced in the image creation to the eye [4].

Different architectures have been proposed to generate waveguide couplers based on holographic recording materials that can be used in AR application [4,5,6]. In general, these couplers are supported on holographic reflection gratings where the spatial resolution required for the material is over 4000 lines/mm. Recently, our group proposed an alternative scheme using transmission holographic elements [7]. The fabrication architecture was tested with three different photopolymers in order to optimize their chemical composition [8]; the first one is based on polyvinyl alcohol acrylamide, PVA/AA, the second one, a nanoparticle-thiol-ene, NPC, and in the last, a penta/hexa-acrylate-based polymer with dispersed nematic liquid crystal molecules. We also proposed three schemes adapted to each material properties.

Currently, there are many photopolymerizable materials, even commercial ones [9], tested for holographic applications [10,11,12]. In general, the photosensitive film can contain one or more monomers, a dye, and a binder to apport a physical resistance. There are many advantages of these materials such as their low price, good optical properties, and the flexibility to change the chemical composition to adapt the material for a particular application. For example, for holographic data storage, high thickness is required so the dye concentration is reduced [13]. In addition, new chemical compounds have been introduced to the classical photopolymers, such as a different kind of nanoparticle; in a photo-polymerization induced phase separation process (PIPS) in which the nanoparticles diffuse to dark zones in the diffraction grating, this effect can provide more structural consistency to the layer, reduce the shrinkage, and increase the refractive index modulation [14,15,16,17]. A type of nanoparticles are the liquid crystal molecules [18]. Holographic polymer-dispersed liquid crystals are known as HPDLCs. In this case, the orientation of the liquid crystal molecules in dark zones can be induced by means of an electric or magnetic field in the case of ferromagnetic liquid crystal [19] and it produces a refractive index variation which changes the diffraction efficiency. Therefore, the grating develops a dynamic behavior that can be modified by means of an electronic device.

In our previous works we found that one of these HPDLC compositions works perfectly to trap light in the waveguide using normal incidence. We expect to measure the tunability of the hologram diffraction efficiency with the electric field. Nevertheless, the samples presented in previous works [8] do not change their properties any electric field is applied. In this work, we deepen in this point checking some chemicals’ composition variation to optimize the wave-couplers response. We consider this point critical for applications such as see-through devices. We studied the recording process, the angular response of the recorded holograms and their tuneability applying an external voltage.

## 2. Materials and Methods

In previous works, the material analyzed [8], HPDLC, presents a weak reaction when an electric field was applied. The monomer used was dipentaerythritol penta-/hexaacrylate (DPHPA), with a refractive index 𝑛 = 1.490. We used the nematic liquid crystal QYPDLC-036 (LC036) from Qingdao QY Liquid Crystal Co., Ltd. Shandong, China (LC). It is a mixture of 4-cyanobiphenyls with alkyl chains of different lengths. It has an ordinary refractive index 𝑛_0_ = 1.520 and a difference between extraordinary and ordinary index Δ𝑛 = 0.250 [20]. The liquid crystal concentration was set at 28 wt%, for solution 0, see Table 1, as the starting point for component optimization and remained practically unchanged during this process. N-vinyl-2-pyrrolidone (NVP) was used as crosslinker, N-phenylglicine (NPG) as radical generator, octanoic acid (OA) as cosolvent [8] and ethyl eosin (YEt) as dye. N-methyl-2- pyrrolidonev (NMP) was used in combination with NVP in order to control overmodulation during hologram recording [8]. The prepolymer solution was made by mixing the components under red light, to which the material is not sensitive. The solution was sonicated in an ultrasonic bath, deposited between glass plates 1 mm thick, and separated using glass microspheres as spacers. The microspheres were provided by Whitehouse Scientific with a thickness between 20 and 30 μm. Firstly, we introduced the YEt with the NMP to solve all the dye in the NMP. Secondly, we added NVP or/and NMP to the solution and keep mixing 5 min with the magnetic stirrer bar spinning. In the third place, we introduced OA and kept stirring for 3 min, then we added to the solution NPG under red light because now the solution was sensitive to the green and blue wavelengths. When the solution was again homogeneous, we added the LC036. Once the solution was transparent, under red light, again we introduced the main monomer DPHPA. Due to the high viscosity of this monomer the solution should be stirred with a stick to obtain a transparent mixture again. From the solution 0 (solt 0) studied in previous papers [8,20], we have not detected any significant variation of the transmission efficiency, TE, under electric field between the two glass substrates. Therefore, for solution 1 (solt 1) we have increased the monomer concentration and LC. Nevertheless, we observed some instabilities, and we increased OA in solution 2 (solt 2) together with a slight reduction of the monomer concentration looking for more stability. In solution 3, (solt 3), we have reduced the monomer quantity little bit more, and we have introduced NMP to reduce the refractive index modulation and achieve more stable layers.

The task of the diffraction grating is to couple the energy of the incident beam, whose propagation vector is denoted by ρ, to the diffracted beam, whose propagation vector is denoted by σ. The diffracted beam must be deflected so that the angle formed with respect to the normal of the glass substrate must be higher than the critical angle, θc, (for the interface glass-air) to accomplish total internal reflection. The recording and read out schemes inside the material are represented in Figure 1 using Ewald’s sphere. Where K is the grating vector and can be obtained easily from the two interfering wave vectors, ρ and σ.

The experimental device is an asymmetric transmission holographic setup; it is represented in Figure 2. A Nd:YAG laser tuned at a wavelength of 532 nm was used to record diffraction gratings by means of continuous laser exposure. The laser beam was split into two secondary beams with an intensity ratio of 2.5:1 using a partial mirror. Due to the cross section of the up arm is in this scale higher than the down one. The diameter of these beams was increased to 1 cm using a spatial filter (SF), collimating lens (L) and diaphragm (D), while spatial filtering was ensured. The working intensity at 532 nm was 2.5 mW/cm^2^ and 1.0 mW/cm^2^ for up and down arms. Slanted gratings of 1700 lines/mm were recorded; to do this, the reference beam formed an angle to the normal of −4.8°, whereas the object beam formed an angle of 68°. We monitored in real time the diffraction grating using red light (λ = 633 nm which the dyes do not absorb) using close to normal incidence (0.3°). After recording, the sample was rotated to record the angular response around the first Bragg condition.

It should be noted that the diffraction efficiency (DE) from a waveguide structure used in our experiment cannot be directly measured because the diffracted beam is trapped inside the glass substrate of the sample at a diffraction angle larger than the critical angle at the boundary between the glass substrate and the air. The diffraction beam goes out from the edge of the glass substrate. For this reason, we evaluated the TE defined as the ratio of the transmitted power to the incident one of our holographic optical elements (HOE), without Fresnel correction at s polarization. We found that the sum of TE and DE in our experiment was close to 0.9. In this way angular responses of the waveguide were measured using transmitted beam, to avoid the movement of the detector to capture the diffracted beam. We rotate the sample in order to obtain the angular response and then we can fit the TE as a function of the outside angle to obtain the value of the effective optical thickness (d), the refractive index modulation (Δn), and the absorption and scattering coefficient (α).

## 3. Results and Discussion

The idea of this section is to compare three characteristics of the different solutions analyzed. First, the transmission efficiency, TE, during recording. Second, the angular response around the brag angle, in this case we can fit the experimental data using coupled-wave theories to obtain the optical thickness of the sample, the absorption and scattering coefficient, and the refractive index modulation. Third, we can study the Bragg angle variation of the transmitted intensity when an electric field is applied.

Figure 3 shows the TE versus time for the four solutions. In general, solution 1, 2 and 3 react faster than solution 0. In all cases there is light guided by the glass substrate. The transmission drops around 0.2 for solutions, 0, 2 and 3. Nevertheless for solutions 1 and 2 more scattering can be detected just looking the sample irradiation with He-Ne laser. Furthermore, these two compositions became solid, few hours after the solution preparation, even when conserved with controlled temperature in the closed bottles. For these two cases also the repeatability of the experiments was far from the solutions 0 and 3.

In order to fit the TE as a function of the readout angle, we have used Kogelnik coupled wave theory [21]. For the case of the transmission grating of Figure 4.

Two important facts that can be observed from Bragg’ detuning are the small deviation from the Bragg’s ideal condition and the attenuation of the grating, refractive index modulation, with depth [22]. The fact that the predicted Bragg angle coincides with the measured one means also that there is no shrinkage or swelling in this case [23,24].

The grating fittings provide interesting information about waveguide couplers recording. The fitted parameters obtained in these cases were for Solt 0: ∆*n* = 0.0074 ± 0.0002, d = 30 ± 2 μm and α = 0.0004 ± 0.0002 μm^−1^; for Solt 1: ∆*n* = 0.0099 ± 0.0002, d = 15 ± 2 μm and α = 0.0096 ± 0.0002 μm^−1^; for Solt 2: ∆*n* = 0.0085 ± 0.0002, d = 18 ± 2 μm and α = 0.0083 ± 0.0002 μm^−1^; for Solt 3: ∆*n* = 0.0053 ± 0.0002, d = 36 ± 2 μm and α = 0.0004 ± 0.0002 μm^−1^. It is important to see the huge values of the absorption and scattering parameter for solutions 1 and 2. This agrees with the low repeatability for this kind of layer, therefore only in the thin samples can some interesting results be obtained. For thickness higher than 20 μm, values of α are higher than 0.01 and this samples are so unstable and not able to store holograms. These two compositions present values of the refractive index modulation higher than the solt 0 and 3. After the recording process the crystals can be detected, even outside of the irradiated zone. This crystallization dramatically affects the transparency of the sample, even outside of the irradiated area.

Once the different waveguide couplers are observed under electric field, we have got the results presented in Figure 5. For solution 0 we do not appreciate any change in the TE under 200 V, for voltages higher than 220 all the samples begin to conduct current. It is interesting to note that for many holograms recorded using solution 1, the scattering disappears when applying this electric field. Nevertheless, the scattering appears again once we shut off our device. For solution 2 we can see some interesting variation of the TE as a function for the applied voltage. However, this change is magnified for the holograms recorded using Solution 3, in this case the TE can be increased from 18% to close to 80%.

The optimal behavior for these tunable couplers is that when applying electric field the diffracted beam can disappear. In this sense, in solution 3 it retains some guided light, even in the maximum of the TE. Therefore, for this application it should be better solution 2, in this case, almost all the energy is transmitted, but the diffraction variation is clearly reduced. It is important to find a compromise between the variation of the diffraction efficiency guided by the glass, the tunable coupler contrast, and the reduction of the remaining guided light when the electric field is applied. Another way to improve this kind of holographic optical element is the reduction in the electric field needed to switch the liquid crystal molecules.

It must also be said that very few studies have been made on the tunable capacities of holographic waveguides. The first work on this direction is a recent work by Diao et al. [25] that demonstrates the tunable behavior of a waveguide based on HPDLC gratings by measuring the output intensity as a function of the applied electric field. Fixing the incident intensity to 374 μW, the output intensity could be modulated from 199 to 9 μW. In our study, the measured quantity was the diffraction efficiency, and the results show a similar behavior of the waveguides to those presented in [25]. Another interesting aspect of the work presented is that good results (high diffraction efficiency) are obtained for gratings with relatively high spatial frequencies (over 1700 lines/mm) using HPDLC, which is a considerable good result taking into account the decrease of performance with respect to the spatial frequency of this kind of device [26]. In order to measure the time response of the device, Diao et al. [25] and us [27] used a square-wave voltage of 1 kHz frequency. After recording the diffraction grating, each HPDLC device is exposed to the electric field. The amplitude of the voltage was increased manually. The samples remain in each voltage for the time necessary until the value of the electric current intensity (I) stabilizes (less than 1 s) [27]. These are not exigent times of response, for instance Peng et al. [28] reported time responses less than 500 ms, but for this kind of application we are not so interested in an extremely rapid response. It must also be said that the device is stable and does not suffer much of hysteresis [29], which is a typical drawback of PDLC composites; this has been demonstrated by examining 100 cycles, finding almost no variation in the response of the device. Finally in Figure 6 a holographic waveguide fabricated with HPDLC is presented.

## 4. Conclusions

We have studied and analyzed three different compositions looking for tunable waveguided couplers. In this sense we have increased the crystal liquid concentration together with the monomer. The solutions 1 and 2 have an unstable behavior presenting high values of scattering, and high values of scattering are undesirable to achieve good image transmission in the see-through applications. In order to achieve more stable photopolymer, we have introduced NMP in solution 3. In this case we have reduced the refractive index modulation, but in this case, we have obtained stable gratings tunable TE from 18% to 80%.

## Figures and Tables

**Figure 1 polymers-13-01858-f001:**
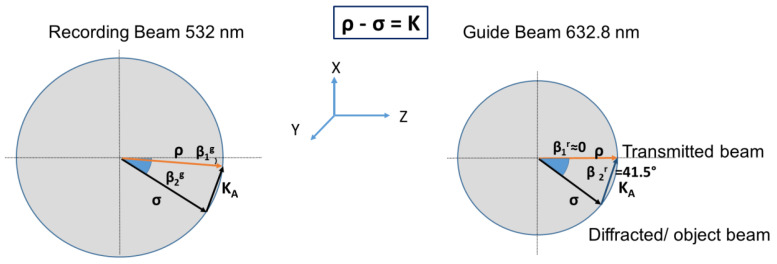
Ewald’s sphere and recording with green light (**left**) and read out with red light (**right**) geometries waveguide coupled element designed for normal incidence of red light.

**Figure 2 polymers-13-01858-f002:**
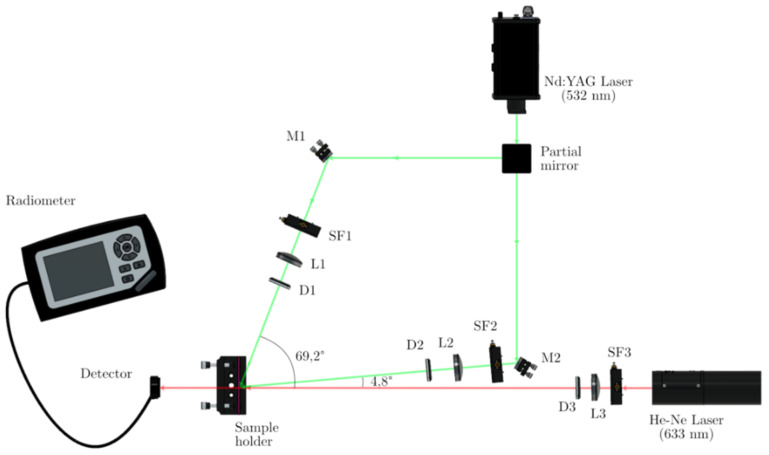
Experimental setup. Partial mirror; Mi: mirror, SFi: spatial filter; Li: lens, Di: diaphragm; radiometer; sample holder, detector and green laser (Nd:YAG) to record the holograms and red laser (He-Ne) to read out.

**Figure 3 polymers-13-01858-f003:**
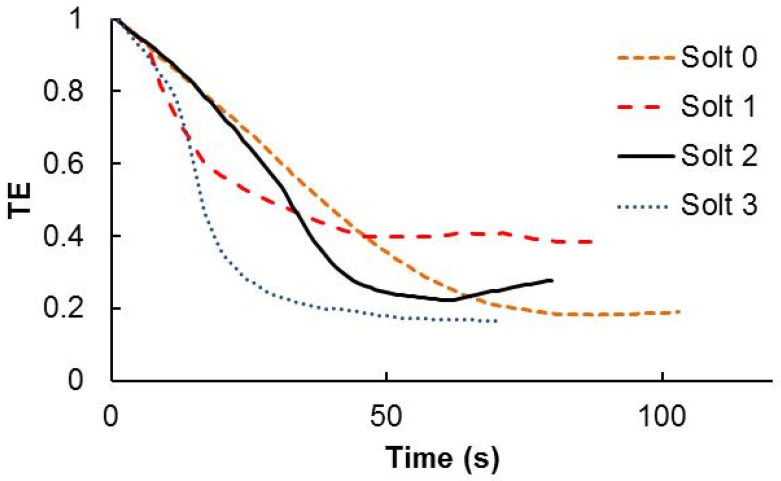
Transmitted efficiency as a function of the exposure time for the different chemical compositions analyzed.

**Figure 4 polymers-13-01858-f004:**
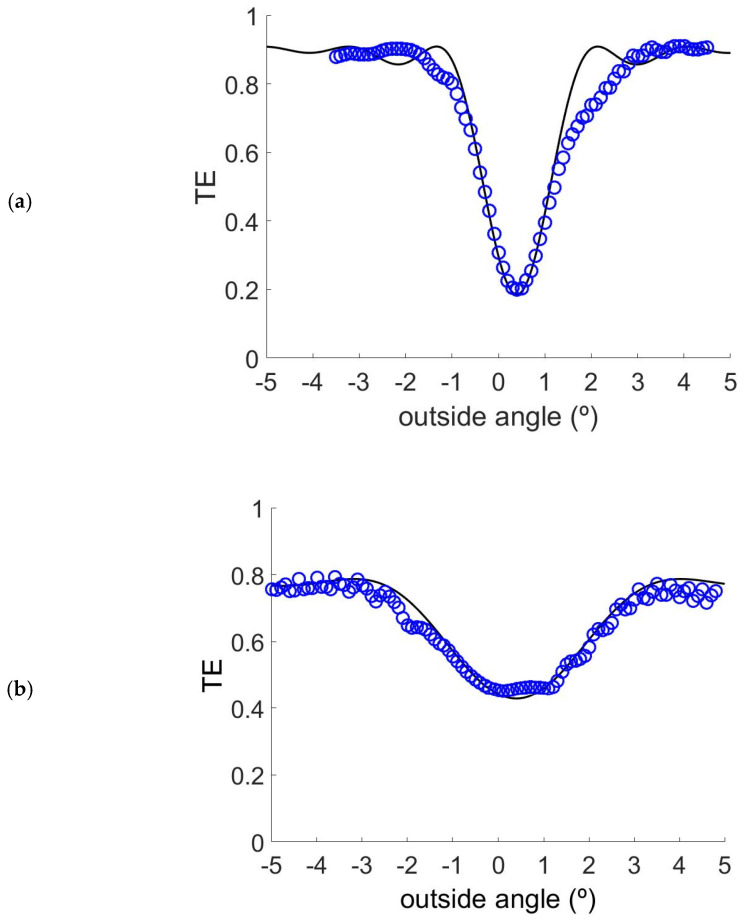
Angular responses for gratings recorded with the different chemical compositions (**a**) Solution 0; (**b**) Solution 1; (**c**) Solution 2 and (**d**) Solution 3.

**Figure 5 polymers-13-01858-f005:**
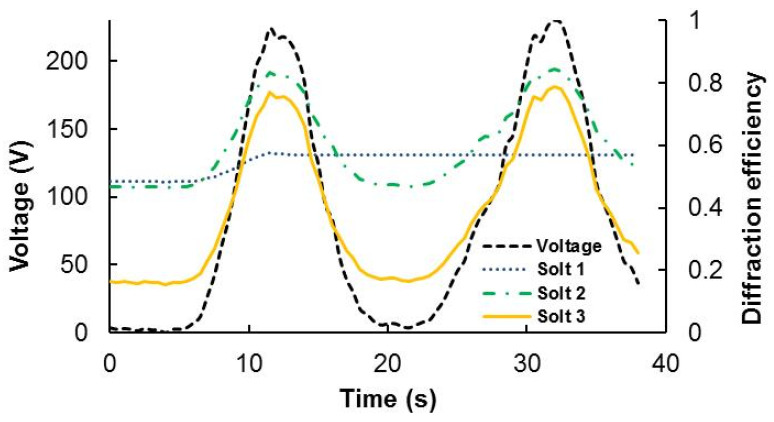
TE dependence from the applied voltage for the different chemical compositions Solt 0; Solt 1; Solt 2 and Solt 3.

**Figure 6 polymers-13-01858-f006:**
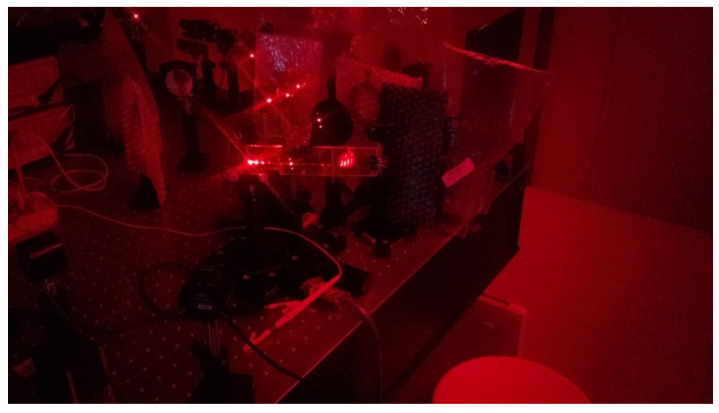
Holographic waveguide under red light illumination.

**Table 1 polymers-13-01858-t001:** Chemical compositions analyzed.

Component	Solution 0	Solution 1	Solution 2	Solution 3
YEt (g)	0.001	0.001	0.001	0.001
NMP (μL)	469	0	0	200
NVP (μL)	0	50	50	50
OA (μL)	144	300	400	300
NPG (g)	0.01	0.01	0.01	0.01
LC036 (μL)	450	750	750	750
DPHPA (g)	1.1	1.4	1.3	1.2

## Data Availability

The data presented in this study are available on request from the corresponding author.

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
