# Peer review of "Tunable Waveguides Couplers Based on HPDLC for See-Through Applications"

_polymers, 2021, doi:10.3390/polym13111858_

Round 1

Reviewer 1 Report

The manuscript presents optimisation of the photopolymer doped with liquid crystals for waveguide coupler application. Response of the photonic structure developed in the doped photopolymer to the electric field was obtained and characterised. Section "Conclusion" contains comprehensive summary of the work and identifies further research direction.

However, the manuscript requires more input to be suitable for publication.

I would recommend introducing the following changes in order to improve the quality of the manuscript.

Major changes:

Unfortunately, the paper has lack of references. The quality of figures and their style have to be improved.  

Minor changes:

Line13

Abbreviation "HPDLC" has to be clarified.

Line 27

References are required

" In particular, photopolymers have been reported as a good system to bring the photons produced in the image creation to the eye.[refs]"

Line 38-40

I would suggest rewording the sentence and add more details about parameters that are under investigation.

Line 61

I would suggest replacing "reacts" with "is sensitive to"

Line 77

Please check symbol for the critical angle. In pdf version I have received the symbol is not correct probably due to format conversion.

Figure 1.

I would suggest including calculations that allows achieving 41.5° as shown on Figure. Also, vector K has to be defined in the text.

Figure 2.

Please check the title of Figure 2 and make sure all components of the set-up are included in the description.

Figure 3-5 have different styles. I would suggest remaking figures to have a single style through the paper.

Line 120.

Pictures of the light guided through the layers made using different compositions as mentioned in Table 1 would be very beneficial for presenting the research results.

Author Response

Dear editor,

Firstly, we would like again to thank the reviewers for their constructive comments and suggestion to improve the manuscript. We have tried to response point-by point. We have marked the changes in the manuscript using yellow highlight.

Reviewer 1

We have introduced more information in the introduction to have a wider view about the different possibilities of photopolymers for photonic applications. Therefore, we have introduced 10 new references.

We have improved the design and quality of figures, especially fig. 3 and Fig. 5.

Minor changes:

We have clarified HPDLC abbreviation in the introduction.

We have introduced new ref. 4 Shen, Z.; Weng, Y.; Zhang, Y.; Wang, C.; Liu, A.; Li, X. Holographic Recording Performance of Acrylate-Based Photopolymer under Different Preparation Conditions for Waveguide Display. Polymers 202113, 936. https://doi.org/10.3390/polym13060936

Line 38-40: We have added the details of the investigation.

We have replaced reacts with is sensitive to.

We have corrected the symbol of the critical angle.

We have introduced K the grating vector.

Figure 2. We have introduced the used components in the description.

We have remake figures 3-5.

With the picture of the light guided it is difficult to appreciate the differences between the compositions.

Reviewer 2 Report

This manuscript reports a partially original research on photopolymer with dispersed liquid crystal molecules with the aim to produce tunable waveguides.

For of all, it is very hard to understand what is the real contribution of this paper to the topic.

The abstract is a general introduction to the field, and it should be rewritten completely to clarify what is the materia under investigation, what are the methods used for the research, what are the main results, adn so on. And what is HPDLC? It is not defined neither in the abstract or in the introduction.

The introduction and the state of the art is very poor. Most of the relevant references are missing, no sufficient references on the chemical and physical aspects of the materials are presented. What is the main scientific problem solved by this contribution and so on.

The materials and preparation procedures are not clear and the text is very confusing.

The esperimental part would need to be rewritten, clarifying the new methods and instruments used here and describing the methodological aspects. Angles and labels in Figure 1 are not defined. The authoers write "The experimental device is an asymmetric transmission holographic setup". What is this setup? Please describe each components and fabrication references. Figures 2 is not clear. In the caption the authors refer to the BS (beamsplitter) but in the figure there is no BS!

The main results of this work should be underlined and discussed in the frame of the literature. Moreover, the fitting models are not described and references are missing.

The overall quality of this paper is very low to be published

Author Response

Dear editor,

Firstly, we would like again to thank the reviewers for their constructive comments and suggestion to improve the manuscript. We have tried to response point-by point. We have marked the changes in the manuscript using yellow highlight.

Reviewer 2

We have improved the introduction to understand the real contribution of the paper.

We have modified the abstract and defined HPDLC in the introduction.

We have amplified the state of the art in the introduction adding 10 new references, including relevant references about chemical and physical aspects of the photopolymers.

The material preparation is described step by step; every reader can prepare our samples and perform our experiments with the appropriate materials.

We have modified the Figure caption of figure 1 and 2 to clarify what it is represented in each case.

The model to fit the angular response of the slanted gratings is the reported by                “Kogelnik, H. Coupled wave theory for thick hologram gratings. Bell Syst. Tech. J. 1969, 48, 2909–2947” and was cited.

The authors have thoroughly revised the document in line with the reviewer and editor suggestions. We are happier with the paper after these constructive positive reviews.

If there are any further concerns, please feel free to contact me.

Yours sincerely,

The authors

Round 2

Reviewer 1 Report

The Authors have provided point-to-point response and introduced most of the changes suggested.

As such, the Authors have included more references in the introduction and expanded it by adding the description of the recent research on photopolymer applications. This makes the introduction more comprehensive and improves the paper in general.

In addition, the Authors have reshaped Figures 3 and 5. But I would also suggest increasing the size of axis titles on Figure 4 to make them more legible. 

In general, I would leave the decision on suitability of the figure style (Figure 3-5) for publication with the Editor as even after introduced modifications the figures have different styles and decrease the quality of the paper presentation.

Regarding the picture of the light guided through the layers made using different compositions as mentioned in Table 1, the Authors claim that it is difficult to appreciate the differences between the compositions. In this case I would recommend including the picture of the composition that has the best performance. In my opinion, the picture would increase the paper presentation and show the scientific soundness of the results discussed.

Author Response

Reviewer 1

We have changed the design of Figure 4, we have increased the size of axis titles to make them more legible.

We have evaluated the possibility to include a picture of the composition that has the best performance, but with the naked eye, all of them looks similar, just using a radiometer we can appreciate the differences on the guided intensity.

Reviewer 2 Report

The authors have only partially answered to the queries. The manuscript still suffers of a low quality of the results and data presentation, figures are not formatted and they seem floating  in the text. Data reported and their relative errors is not correct (look at data reported from line 168-172, significative numbers are not coherent with the standard deviation). Discussion of results is very poor and it is not related to the literature state of the art. Some parts in the Introduction should be rewritten since thay are not clear.

Author Response

Reviewer 2

We have corrected the errors of the refractive index modulation extracted from the fittings from line 168-172, we have marked these changes in yellow.

Round 3

Reviewer 2 Report

Few modifications have been provided and no improvement in the discussion has been added. The quality of the manuscript is low.

Author Response

We are really grateful with reviewer’s comments. Thanks to her/his advices the article has been improved.
Here I explain the changes made in the manuscript:

We have added Figure 6 with a photograph of a waveguide fabricated in our lab.

We have added the following text (with 4 new references) just before the conlusions:

It must also being said that very few studies have been made on the tunable capacities of holographic waveguides. The first work on this direction is a recent work by Diao et al. [25] that demonstrates the tunable behavior of a waveguide based on HPDLC gratings by measuring the output intensity as a function of the applied electric field. Fixing the incident intensity to 374 μW, the output intensity could be modulated from 199 to 9 μW. In our study the measured quantity was the diffraction efficiency and the results show a similar behavior of the waveguides to those presented in [25]. Another interesting aspect of the work presented is that good results (high diffraction efficiency) are obtained for gratings with relatively high spatial frequencies (over 1700 lines/mm) using HPDLC, which is a considerable good result taking into account the decrease of performance  with respect to the spatial frequency of this kind of devices[26].  In order to measure the time response of the device Diao et al.[25] and us [27] used a square-wave voltage of 1 kHz frequency. After recording the diffraction grating, each HPDLC device is exposed to the electric field. The amplitude of the voltage was increased  manually. The samples remain in each voltage for the time necessary until the value of the electric current intensity (I) stabilizes (less than 1 s) [27]. These are not exigent times of response, for instance Peng et al. [28] reported time responses less than 500 ms, but for this kind of application we are not so interested in an extremely rapid response. It must also be said that the device is stable and doesn’t suffer much of Hysteresis [29], which is a typical drawback of PDLC composites; this has been demonstrated by examining 100 cycles, finding almost no variation in the response of the device. Finally in Figure 6 a waveguide fabricated is presented.